# Organic Small-Molecule Electrodes: Emerging Organic Composite Materials in Supercapacitors for Efficient Energy Storage

**DOI:** 10.3390/molecules27227692

**Published:** 2022-11-09

**Authors:** Yuanyuan He, Qiaoqiao Wei, Ning An, Congcong Meng, Zhongai Hu

**Affiliations:** 1College of Chemistry and Chemical Engineering, Northwest Normal University, Lanzhou 730070, China; 2College of Chemistry and Chemical Engineering, Lanzhou Jiaotong University, Lanzhou 730070, China; 3School of Electronic and Information Engineering, Lanzhou City University, Lanzhou 730070, China

**Keywords:** organic small-molecule electrodes, energy storage, supercapacitors, redox activity

## Abstract

Organic small molecules with electrochemically active and reversible redox groups are excellent candidates for energy storage systems due to their abundant natural origin and design flexibility. However, their practical application is generally limited by inherent electrical insulating properties and high solubility. To achieve both high energy density and power density, organic small molecules are usually immobilized on the surface of a carbon substrate with a high specific surface area and excellent electrical conductivity through non-covalent interactions or chemical bonds. The resulting composite materials are called organic small-molecule electrodes (OMEs). The redox reaction of OMEs occurs near the surface with fast kinetic and higher utilization compared to storing charge through diffusion-limited Faraday reactions. In the past decade, our research group has developed a large number of novel OMEs with different connections or molecular skeletons. This paper introduces the latest development of OMEs for efficient energy storage. Furthermore, we focus on the design motivation, structural advantages, charge storage mechanism, and various electrode parameters of OMEs. With small organic molecules as the active center, OMEs can significantly improve the energy density at low molecular weight through proton-coupled electron transfer, which is not limited by lattice size. Finally, we outline possible trends in the rational design of OMEs toward high-performance supercapacitors.

## 1. Introduction

Due to the severe resource depletion and environmental pollution caused by the over-exploitation of fossil fuels, sustainable energy development has received extensive attention—in particular, energy storage devices, including electrochemical supercapacitors (SCs) and rechargeable batteries [1,2,3,4,5,6]. Among them, SCs, as a new energy storage device between traditional capacitors and chemical power devices, have the characteristics of good stability, a long cycle life, and environmental protection performance [7,8,9]. Although the energy density of SCs is lower than that of batteries, they can instantly retain and provide a large amount of charge, thus demonstrating higher power density [10,11]. Therefore, building and developing supercapacitors with high energy density takes considerable effort while ensuring high power density. Various carbonaceous materials and metal oxides have been widely used as electrodes with high power density, a long life, and large rate capacity [12]. In particular, inorganic transition metal materials show excellent electrochemical properties [13,14]. For example, Du and colleagues obtained a 3D honeycomb nitrogen-doped porous carbon network (N-CN/Co_3_O_4_) modified by nano-cobalt [15]. The unique synergistic effect and the introduction of nitrogen enabled the material to achieve a specific capacitance of 1115.4 F g^−1^ at 1.0 A g^−1^. In addition, Wu et al. further promoted the total exposure of active sites in hierarchical NiS/carbon hexahedrons by effectively adjusting the structure and morphology [16]. The optimized NiS/carbon electrode (NiS/NTA-2) achieves the expected superior energy storage performance, and the specific capacitance is 1530.4 F g^−1^ at 1.0 A g^−1^. However, traditional inorganic materials rely on scarce and non-renewable resources. Therefore, developing innovative, green, and high-performance electrode materials is the key to the further development of supercapacitor materials.

Organic materials provide an excellent opportunity to improve the existing energy storage technology further, and their reversible redox process shows potential high specific capacitance and high energy density. The study of organic molecules in supercapacitors can be traced back to the electrochemical reduction of 2-nitro-1-naphthol in 2004 [17]. The obtained electrode provides additional pseudocapacitance and high electric double-layer capacitance. However, the poor cyclic stability of individual small organic molecules in the charge/discharge process limits its more comprehensive application. In 2008, Pickup et al. reported that carbon fabric anode materials modified by anthraquinone were assembled into supercapacitors with high energy and power density [18]. On this basis, in 2011, Hu’s team fixed tert-butylhydroquinone on graphene sheets by π–π stack [19]. Since then, several research groups have tested various small organic molecules, such as anthraquinone derivatives [20], catechol derivatives [21], sulfanilic acid azo-cromotrop [22], methyl green [23], and monoamino phthalocyanine [24]. In 2019, the concept of organic small-molecule electrodes (OMEs) was proposed for the first time in the study of Hu et al. [25]. The composite materials obtained by fixing redox-active organic small molecules on a conductive carbon matrix are called OMEs. From the electrochemical characteristics and the corresponding kinetic analysis, for inorganic battery-type materials, most of the charges are stored by the Faraday reaction at constant potential [26]. In the process of electrolyte ion embedding and detachment, bulk phase diffusion will lead to lattice expansion/contraction affecting the electrode material’s rate performance and power density. [27,28,29,30,31,32]. On the contrary, OMEs store charge by reversible redox reactions near the surface of a conductive substrate. Although redox reactions occur in both OMEs and inorganic battery materials, a significant difference is that OMEs are not limited by the sluggish reaction kinetics of solid-state diffusion [33,34,35]. In the cyclic voltammetry (CV) curve, most OMEs showed well-defined oxidation/reduction peaks without phase changes [36,37,38,39,40]. Further, the conductive carbon substrate in OMEs can also contribute to the electric double-layer capacitance through electrostatic storage [41,42,43]. In addition to the unique energy storage mechanism, OMEs are designed to meet the construction principles of a green, all-carbon energy storage device. The organic small molecules used in OMEs are usually composed of light elements, which have flexibility, easy availability of raw materials, and environmentally friendly [44]. As we all know [45]:(1)$$C\left(F\;g^{-1}\right)=Q/V=\left(\frac{\mathrm{nF}}{m}\right)X/V.$$

The m is the molecular weight of the active material, n is the number of electrons, and F is the Faraday constant. The theoretical capacitance of OMEs mainly depends on the number of active sites per weight [46]. Therefore, multi-electron Faraday reactions occurring at low molecular weights are the key to excellent electrochemical properties of OMEs.

Our group’s research about OMEs in the past decade found that the diversity of OMEs mainly stems from the variation of active center functional groups or different connection modes between organic small molecules, and conductive carbon substrates. The adjustable structure provides an opportunity to prepare electrode materials with a high potential difference, which is beneficial for obtaining excellent energy density. When the electron cloud around the redox center of organic small molecules is attracted/repelled by functional groups, the working potential shifts to the direction of positive/negative. In addition, due to the electrocatalysis of carbon substrates, OMEs can obtain excellent power density while enhancing the pseudocapacitance response.

This review will summarize the latest progress of OMEs as an efficient energy storage electrode material from three aspects: material design, classification, and synthesis strategy. It includes the design motivation, structural advantages, charge storage mechanism, and various electrode parameters of OMEs. Furthermore, according to the classification of reversible electrochemically active small organic molecules, we discuss the redox processes of OME materials in detail (Figure 1). Finally, we outline possible trends in the rational design of OMEs toward high-performance supercapacitors.

## 2. Design of Organic Small-Molecule Electrodes

Over the past decade, hundreds of research articles on OMEs have been published. As shown in Figure 2, a considerable amount of work has allowed us to answer the inevitable questions that have been haunting the minds of energy storage researchers:

### 2.1. What Is the Design Motivation of OMEs?

According to the energy storage mechanism, electrode materials can be divided into electrostatic double-layer capacitance (EDLC) and pseudocapacitive (PC) materials. Ideal EDLC achieve uniform charge storage across the entire potential range by forming an electric double layer at the electrode/electrolyte interface [47]. For pseudocapacitors or hybrid supercapacitors, the electrical energy is not only stored by electrostatic double-layers but also concentrated in redox reactions (or intercalation) [48,49]. Furthermore, due to the high-speed kinetic of the Faradaic redox reaction, the material is not limited by semi-infinite diffusion [50]. Traditional pseudocapacitive electrode materials include transition metal oxides (RuO_2_, MnO_2_, and Fe_3_O_4_), transition metal sulfides (MoS_2_), and conductive polymers, which store charge through surface intercalation/deintercalation of electrolyte cations/protons. In particular, although inorganic metal materials exhibit excellent capacitive performance, the high cost and volume expansion/contraction during long-term cycling hinder their application in large-scale energy storage systems [51]. On the contrary, organic compound materials composed of light atoms offer an opportunity to develop high-performance, green, and sustainable supercapacitor energy storage devices. Despite abundant sources, tunable redox activity at the molecular level, and environmental friendliness, organic molecular materials still face some challenges in applications, such as low electrical conductivity and dissolution in aqueous electrolytes that may affect electrode power performance and cycle stability. To achieve high-performance and green energy storage, organic compounds with redox activity can be immobilized on conductive carbon substrates through non-covalent interactions or chemical bonds. The resulting composites are called organic small-molecule electrodes. This strategy can further prevent the energy loss caused by the shuttle effect, reviving the research on organic electrode materials [52].

### 2.2. What Are the Advantages of OMEs?

In terms of working mechanism:

First, OMEs are excellent candidates for next-generation energy storage systems. The impressive performance mainly derives from pseudocapacitance and electric double-layer capacitance. In particular, organic small molecules are regarded as active centers, and the charge storage mechanism is based on redox reactions of specific functional groups (e.g., C=N, N=N and C=O bonds). This surface-controlled behavior operates at a rate comparable to the rate of electric double-layer charge storage in capacitive materials [24]. In contrast to conventional pseudocapacitive materials that rely on changes in metal oxidation state to store charge, OMEs are not limited by lattice size and provide additional Faraday pseudocapacitance. Second, the electrochemical catalysis of carbon substrates enables OMEs to display fast kinetic properties. The electrocatalysis of carbon substrate means that the porous composite structure formed by conducting carbon substrate and organic molecules can provide a larger contact area for the electrolyte, thus further contributing a larger double-layer capacitance. For the OME materials modified with organic small molecules, the integral area of the CV curve is often larger than that of the corresponding substrate materials. It can be confirmed that organic molecules and conductive substrates have a positive influence on each other. On the one hand, the organic molecules immobilized on the OMEs can act as spacers to suppress the aggregation of conductive carbon substrates, enabling fast mass transfer and efficient utilization of active sites. On the other hand, in the previous study of OMEs, the best adsorption direction of organic molecules on a conductive carbon substrate is obtained by DFT calculation [25]. Some small organic molecules will form directionally parallel conjugated sp2 networks between conductive carbon substrates. Therefore, the degree of disorder and defects of the conductive substrate is repaired. The unique structural orientation can reduce the polarization process of electrochemical reactions and accelerate the reaction kinetics.

In terms of structure:

Since individual organic small molecules show low electrical conductivity and solubility, they are anchored on carbon-based materials to prepare highly conductive OMEs. Covalent bonding can be used to prepare stably connected OME materials, and non-covalent methods can also be selected to ensure capacitive performance without significantly changing the π-conjugated system of the substrate. Because OME materials span a wide range of organic compounds, changing the active center’s functional group or adjusting the backbone conjugation degree can realize the flexible design of OMEs at the molecular level. In particular, organic compounds with low molecular weights and multiple redox moieties can provide high specific capacitance. In addition, the carbon framework in OMEs can also provide high electrical conductivity, interconnected porous structure, large specific surface area, and excellent mechanical properties.

### 2.3. How Do the OME’s Electrode Performance Parameters Compare with Other Emerging Materials?

In general, the potential at which redox processes occur is an important parameter affecting the energy density of electrodes. When the potential difference between the redox current response of the positive and negative electrodes is more considerable, the platform position of the two-electrode system is higher [15]. For OMEs, the redox potential usually determines by the species of redox centers in organics. Further, it can regulate the working voltage through the participation of an electron-donating group or electron withdrawing group. The electron affinity increases when the electron cloud around the redox center is attracted. Since it is more challenging to extract electrons from the redox center, the redox potential shifts in a more positive direction and vice versa [53]. Therefore, according to the needs, we can select suitable OME materials to achieve the expected potential difference between the positive and negative electrodes, which is beneficial to obtain a larger energy density.

The low cyclability of organic compounds often attributes to their dissolution in electrolytes. Instead, combining organic small molecules and conductive carbon frameworks in OMEs can break this bottleneck. In addition, the high electrical conductivity, interconnected porous structure, large specific surface area, and excellent mechanical properties of the OMEs can provide fast ion transport paths and low charge transfer resistance. Apparently, the organic compounds immobilized on the conductive substrate can show satisfactory rate performance and cycling stability at high current densities.

## 3. Classification of Organic Small-Molecule Electrodes

Organic small-molecule materials with electrochemically active and reversible redox groups are mainly of three categories: carbonyl compounds (quinone derivatives, ketones, amides, carboxylic acids, and anhydrides), nitrogen-containing heterocyclic compounds, and sulfur-containing compounds. Due to their different molecular structures, they exhibit significantly different behaviors regarding capacitance, voltage, rate capability, cycling stability, and fabrication complexity. This section will discuss these organic small-molecule electrodes in detail (Figure 3). Table 1 summarizes the detailed electrochemical properties of the organic molecule electrode in different types, which display the configuration, electrolyte, capacitance, energy, and power density.

### 3.1. Carbonyl Compounds

#### 3.1.1. Quinone Derivatives

Quinones contain two adjacent/separate carbonyl groups in an unsaturated six-membered ring structure [70,71]. Therefore, the redox reaction of quinone and its derivatives involves the transfer of two electrons and two protons. In carbonyl (C=O) compounds, quinones and ketones have higher redox potentials [72]. In particular, quinone-decorated highly conductive organic molecule electrodes exhibit high capacitance and stable performance, which can be used in pseudocapacitors. Our recent progress in the synthesis of anthraquinone-based OMEs will be briefly mentioned.

Our groups demonstrated a strategy to synthesize anthraquinone (AQ)-functionalized graphene with interconnected 3D macroporous structures via π–π stacking (Figure 4a) [54]. The obtained AQ/GF showed reversible electrochemical behavior in CV curves. The redox reaction caused by proton-coupled electron transfer of the quinone carbonyl group shows that the AQ anode peak is approximately −0.087 V and the cathode peak is approximately −0.125 V (Figure 4a). AQ/GF delivered a gravimetric capacitance of 396 F g^−1^ at 1 A g^−1^ using 1 M H_2_SO_4_ as an electrolyte. Further, it exhibited no significant degradation upon 2000 charge/discharge cycles. Based on the good specific capacitance and excellent stability of anthraquinones in acidic electrolytes, we developed a series of quinone-based OMEs with different electrochemical properties and redox peak positions by adjusting the structural design and synthesis conditions. The benz[a]anthracene-7,12-quinone (BAQ) were immobilized on reduced graphene oxide to form porous three-dimensional aerogels (Figure 4b) [55]. Among them, the BAQ/rGO-x composite electrode has a prominent redox peak at 0.055/−0.081 V in the potential window from −0.4 to 0.6 V. Compared with the bare reduced graphene oxide electrode, the specific capacitance of BAQ/rGO is significantly improved (400.3 F g^−1^ at 1 A g^−1^) (Figure 4b). To increase the number of active sites (quinone carbonyl groups), we further selected 5,7,12,14-pentacenetetrone (PT) as the guest molecule, which large π-conjugated backbone consists of two redox anthraquinones coupling [56]. There is a pair of clear peaks centered at 0.162/0.092 V, confirming the multi-electron redox reaction of PT-0.5@RGO. The PT-0.5@RGO electrode delivered a specific capacitance of 433.2 F g^−1^ at 5 mV s^−1^ and could retain a capacitance above 377.4 F g^−1^ even at a scan rate of 100 mV s^−1^. It can be seen that the expanded conjugated polycyclic quinone derivatives can enable four carbonyl groups to simultaneously participate in the redox process, which plays a more important role in improving the capacitive performance. In conclusion, when the conjugate system of quinone derivatives increases, the redox peak of the organic small-molecule electrode moves to the positive region. Modifying the number of carbonyl groups in the bone could improve the electrochemical performance, providing a new way to design high-performance organic molecule electrodes. Moreover, 1-hydroxyanthraquinone (HAQ) has been adsorbed onto dissected carbon nanotubes (rDCNTs) through noncovalent interaction (Figure 4c) [57]. Figure 4c shows HAQ-rDCNTs with a pair of well-defined reversible redox peaks of approximately −0.15 to −0.1 V. Since the electron-donating group (hydroxyl) reduces the electron affinity of the compound, the redox potential is low compared to that of anthraquinone. In the three-electrode configuration, functionalized rDCNTs with the 1-hydroxyanthraquinone molecular have an enhanced specific capacitance of 324 F g^−1^ at 1 A g^−1^ compared to unfunctionalized carbon nanotubes.

Notably, the anthraquinone derivative alizarin (AZ) immobilizes on 3D self-assembled graphene hydrogel (SGH) [6]. Due to redox reactions and fast kinetic properties, AZ-SGHs show specific capacitances up to 350 F g^−1^ (1 A g^−1^). Among them, the functional groups (hydroxyl and carbonyl) of AZ provide two multi-electron redox centers. Therefore, two pairs of symmetrically reversible peaks at +0.83 V and −0.18 V can be observed on the CV curve. When assembling symmetric supercapacitors (SSCs) with AZ-SGH electrodes, as the cell voltage increases from 0 V, the potentials will move positively and negatively from relative values, respectively. As shown in Figure 5, in the initial charging stage, there is no obvious redox peak on the CV curves of the positive and negative electrodes. At this point, the component stores charge through the electric double layer. When the voltage reaches 1.02 V, the oxidation reaction of the O_2_/R_2_ pair occurs at the positive electrode and the reduction reaction of the O_1_/R_1_ pair occurs at the negative electrode. After the superposition of the two half-reactions, a pair of reversible redox peaks at 1.02 V appeared in the AZ-SGHs SSC. Benefiting from these two pairs of mirror-symmetric redox peaks, the AZ-SGHs SSCs exhibit good self-synergistic and potential self-matching behaviors. Furthermore, indanthrone (IDT), as an anthraquinone derivative, self-assembles with RGO under a donor–acceptor (D–A) interaction strategy [58]. Two pairs of reversible redox peaks in resulting RGO–π–IDT heterojunctions can clearly be observed from the cyclic voltammetry curves. Benefiting from the very smooth electron transfer provided by the unique D–A interaction, the RGO–π–IDT heterojunctions reported a superior gravimetric and volumetric capacitance of 535.5 F g^−1^ and 685.4 F cm^−3^.

#### 3.1.2. Ketones

Apart from anthraquinone derivatives, ketones can also provide pseudocapacitive properties to undergo redox reactions. As early as 2013, Volker Presser et al. used a facile method to modify 1,4-naphthoquinone, 9,10-phenanthrenequinone, and 4,5-pyrenedione on carbon onions as organic small-molecule electrodes [59]. Among them, the PY-OLC not only delivers an electrode capacitance of 264 F g-1 at a current density of 1.3 A g^−1^ but also displays excellent rate performance and a long cycle life with a capacity retention of 97 % after 10,000 cycles. Additionally, due to the presence of carbonyl and aromatic amine groups in the backbone, 2-aminopyrene-3,4,9,10-tetraone (PYT-NH_2_) is regarded as an excellent candidate with a fast redox rate and high theoretical pseudocapacitance. Zhongyi Liu’s group prepared a PYT-NH_2_/RGO organic small-molecule electrode by covalent bonding [60]. This quinoid oxygen grafted on the surface of the carbon material leads to the superior specific capacitance of 326 F g^−1^, which is much higher than that of pristine PYT-NH_2_. Asymmetric devices were fabricated using activated carbon (AC) and PYT-NH_2_/RGO. When the energy density is 15.4 Wh kg^−1^, the maximum power density is 6078.8 W kg^−1^.

#### 3.1.3. Amide

Although quinone-based and ketone-based OMEs provide certain pseudocapacitive properties, nitrogen atoms with lone electron pairs in amide molecules can further improve redox activity [46]. Ruiguang Xing et al. prepared a novel functionalized reduced graphene oxide (rDGO) using *N*-(4-aminophenyl)-3-oxobutanamide [61]. Subsequently, MnO_2_ nanoparticles were deposited on the surface of rDGO as a high-performance composite supercapacitor electrode material (DGM4). The DGM4 electrodes showed a specific capacitance of 267.4 F g^−1^ at 0.5 A g^−1^ while maintaining high cycling stability with 97.7% of its initial capacitance after 1000 cycles. Another method to prepare acid amide-based OMEs is bonding -COOF on conductive substrates with -NH_2_ on organic small molecules. In the study by our groups, 2-amino-3-chloro-1,4-naphthoquinone (2-NTQ) molecule was covalently modified on the surface of RGO by amidation [36]. The electrochemical test results indicated that resulting 2-NTQ-RGOs have capacitances of 453 F g^−1^ at 1 A g^−1^ and cycling stability (83% capacitance retention over 8000 cycles).

#### 3.1.4. Carboxylic Acids and Anhydrides

Anhydride-based materials usually have one aromatic and two anhydride groups, such as 3,4,9,10-perylene-tetracarboxylicacid-dianhydride (PTCDA), for the preparation of self-supporting OME membranes with fluffy and porous structures [62]. Two pairs of redox peaks clearly appear in CV curve, caused by a typical chemical bond reaction between the two conjugated anhydride groups on PTCDA and lithium ions in the electrolyte. The excellent ion transport properties of the rGO framework and the fast Faradaic reaction of the PTCDA molecule provide a gravimetric capacitance of 242.9 F g^−1^ at 2 A g^−1^. In addition, carboxylic acid-based organic small molecules can also be functionalized to combine with conductive substrates to prepare OMEs with excellent energy storage properties. Mohammad Mazloum-Ardakani et al. reported an arginine amino acid (Arg) decorated graphene oxide (Arg/GO) supercapacitor [63]. Arginine can act as a Faraday reaction source and improve the hydrophilic polar site of the electrode. The faster ionic transitions lead to a higher specific capacitance (532 F g^−1^) of Arg/GO than unmodified GO. We obtained PTCA/RGO1 by using non-covalent functionalization of RGO1 with 3,4,9,10-perylenetetracarboxylic acid (PTCA) [64]. Based on the unique electrochemical reaction of ultrafast protonation/deprotonation of carboxyl groups in PTCA, PTCA/RGO1 exhibits a high specific capacitance value of 422.7 F g^−1^ at 10 mV s^−1^. Asymmetric supercapacitors assembled with PTCA/rGO1 and PANI/rGO2 have a large voltage window (1.6 V) and show an energy density of 14 W h kg^−1^ at a power density of 520 W kg^−1^.

### 3.2. Nitrogen-Containing Heterocyclic Compounds

As with carbonyl compounds and amide-based organic molecule electrodes, nitrogen-containing heterocyclic compounds can also be combined with conductive substrates. For example, Nageh K. Allam et al. prepared functionalized graphene sponge (FG) by a nucleophilic addition reaction between adenine-NH_2_ groups and carboxylic acid/epoxy groups on graphene oxide (Figure 6a) [65]. Figure 6b,c show that the interconnected and porous three-dimensional wrinkled network provides better contact interfaces and facilitates charge transfer kinetics. Due to the presence of nitrogen atoms in adenine, the SFG electrode exhibits a weak cathodic peak at ~0.5 V and a corresponding anodic peak at ~0.32 V in Figure 6d. The SFG electrodes showed a maximum specific capacitance of 333 F g^−1^ at a scan rate of 1mV s^−1^ and exhibited excellent cycling retention of 102% after 1000 cycles at 200 mV s^−1^ (Figure 6e). Yong Zhang’s group demonstrated a facile method to prepare graphene hydrogels (APGHs) with abundant heteroatom functional groups (N, O) using 2-aminopyridine [66]. In particular, due to the high specific surface area and three-dimensional interconnection structure of APGHs-40, the assembled device can achieve a specific capacitance of 266.7 F g^−1^ at 0.3 A g^−1^. This symmetric supercapacitor delivered a high energy density of 9.3 Wh kg^−1^ and a power density of 75.0 W kg^−1^. In addition, APGH-40-based device displayed outstanding capacitance retention of 102.9% after 10,000 cycles. Additionally, 7-aminoindole was used as a pseudocapacitive compound to prepare graphene-based electrodes [67]. The as-synthesized 7-AirGO composite exhibits a high capacitance of 425.73 F g^−1^ at 0.5 A g^−1^, which is a significant improvement compared to the reduced graphene oxide electrode and the 7-Ai electrode.

### 3.3. Sulfur-Containing Compounds

Benzo[1,2-b:4,5-b’]dithiophene-4,8-dione (BDTD) is a large planar molecule with a fused heteroaromatic structure [68]. It can be adsorbed on conductive reduced graphene oxide (rGO) via π–π interactions to form three-dimensional interconnected functionalized gels (BDTD-rGO). As a result, BDTD-rGO displayed a highly reversible specific capacity of 360 F g^−1^ at 1 A g^−1^ and remained at 96.4% of its initial capacity after 1000 cycles at 5 A g^−1^. Sulfur-containing compounds can act not only as sulfur dopants but also as reductants and modifiers. For example, in the research of Kelei Zhuo’s group, thiosalicylic acid (TSA) was hydrothermally treated on graphene oxide precursor to prepare thiosalicylic acid-modified graphene aerogel (TGA) [69]. The symmetric supercapacitor assembled with TGA as the electrode material can provide a specific capacitance of 236 F g^−1^ at 1 A g^−1^. The results show that the doping of S atoms provides additional active sites for the OMEs and accelerates the charge transfer at the electrode/electrolyte interface.

## 4. Preparation Strategies of Organic Small-Molecule Electrodes

OMEs are composed of organic small molecules and conductive carbon substrates. Usually, non-covalent and covalent functionalization strategies are used to prepare OMEs. As summarized in Table 2, various OMEs can be synthesized based on covalent bonding and non-covalent interactions.

### 4.1. Covalent Functionalization

The covalent functionalization of the conductive substrate by organic small molecules is accompanied by the rehybridization of C atoms from sp^2^ to sp^3^ [92]. Typical reaction types are mainly divided into epoxy ring-opening reactions, the addition of diazonium species, acylation reactions, and cycloaddition reactions. In addition, heteroatoms can also be doped on the surface of conductive substrates in covalent bonds to generate new properties.

#### 4.1.1. Epoxide Ring-Opening Reactions

Under the action of nucleophiles, the ring-opening reaction can covalently bond various small organic molecules onto a conductive substrate, such as aliphatic/aromatic amines (Figure 1).

Chen et al.’s recently prepared RGO@PPD films confirmed that p-phenylenediamine (PPD) could react with epoxy functional groups on graphene oxide surfaces [73]. The resulting C-N groups can interact with anions in the discharge state, effectively enhancing the pseudocapacitive contribution of the material. In addition, the PPD molecules act as intercalation to relieve the agglomeration of the graphene sheets, providing an unimpeded channel for electrolyte ion transport in the electrode. At the same time, the formation of covalent bonds can reduce the solubility of PPD and ensure the high stability of hybrid supercapacitors. Benefiting from the above advantages, the assembled supercapacitor exhibits a remarkable areal capacitance of 3012.5 mF cm^−2^. In particular, it has a high energy density of 1.1 mWh cm^−2^ at a power density of 0.8 mW cm^−2^.

Similarly, as shown in Figure 7a, we achieved large-scale anchoring of active organic PPD molecules by utilizing large oxygen-containing functional groups on the sidewalls of dissected carbon nanotubes (DCNTs) [74]. Because PPD acts as a molecular pillar, PPD-C-DCNTs show loose nanotube microstructure in Figure 7b. PPD-C-DCNTs showed the highest capacitance response and two pairs of weak Faradaic redox peaks in the CV curve (Figure 7c). Covalently functionalized DCNT (PPD-C-DCNT) electrode material delivered a gravimetric capacitance of 388 F g^−1^ at 1 A g^−1^ using 1 M H_2_SO_4_ as an electrolyte. Benefiting from the interaction between the epoxy groups of graphene oxide and aniline, Yang et al. used a simple one-step hydrothermal to synthesize DQ-RGO, which covalently modified 2,6-diaminoquinone (DQ) molecules onto the GO through epoxy ring-opening reaction (Figure 7b) [75]. The experimental results show that the electroactive anthraquinone subunits exhibited a reversible two-electron transfer process. Under the effective combination of the Faraday pseudocapacitance and the double-layer capacitance, DQ-RGO exhibits an excellent specific capacitance (332 F g^−1^ at 1 A g^−1^) in an acidic electrolyte solution.

#### 4.1.2. Diazotisation

A well-established and highly effective modification for carbon supports (graphene, carbon nanotubes, etc.) is functionalization using aryldiazonium salts, which is accompanied by the rehybridization of carbon atoms from sp^2^ to sp^3^ [93]. As shown in Figure 2, the mechanism involves the transfer of a delocalized electron from the carbon surface to the diazonium cation, forming an aryl radical tightly bound to the substrate [94]. Subsequently, the radicals are coupled to the substrate’s sp^2^ hybridized carbon lattice atoms to obtain strong covalent bonds.

Ensafi’s group investigated the modification of reduced graphene oxide (RGO) with thionine (Th) to synthesize Th/RGO composites via a diazonium reaction (Figure 8a) [76]. They proved that RGO with high surface area and Th with electrochemical activity significantly enhanced the specific capacitance. The electrochemical performance of Th/RGO was evaluated in 0.5 M H_2_SO_4_ aqueous solution using three-electrode system. The maximum specific capacitance of the diazotized product (Th/RGO) is 1255 F g^−1^ at a current density of 0.5 A g^−1^. Subsequently, the same design strategy was also adopted by Ensafi et al. to enhance the electron transfer of organic small-molecule electrodes for supercapacitor [77]. The as-prepared adenine@rGO nanocomposite exhibited remarkable performance with a specific capacitance as high as 700 F g^−1^ at the current density of 0.5 A g^−1^. In 2022, Bradley Easton et al. covalently embedded phenazine iron adducts on carbon supports through a diazonium coupling reaction and then added FeCl_3_ to coordinate nitrogen−modified carbon supports [78]. In the three-electrode system with 0.5 M H_2_SO_4_ as the electrolyte, the capacitance of BPphenazin-Fe700 was further substantially increased to 287 F g^−1^. The above results clearly demonstrate the importance of pseudocapacitive surface functional groups (phenazine and iron) for carbon materials’ increased charge storage ability.

Our groups selected 2-amino-3-chloro-1,4-naphthoquinone (ACNQ) as their research object and covalently grafted it onto graphene nanosheets (GNS) using aryldiazonium salts [79]. When using 1 M H_2_SO_4_ as the electrolyte, the ACNQ-functionalized GNS (CNQ-GNS) electrode delivered exceptionally high gravimetric capacitances of 364.2 F g^−1^ at a discharge rate of 1 A g^−1^, much higher than that of bare GNS (190 F g^−1^). Notably, such promising performance was attributed to the combination of faradaic capacitance originating from the organic functional groups covalently bonded on the GNS and double-layer capacitance of highly conductive GNS scaffold (Figure 8b).

In 2021, we proposed a strategy to covalently graft 2,6-diaminoanthraquinone on GO surface to form strong carbon–carbon bonds between nitrogen ligands and the carbon surface [95]. The covalently-pillared graphene network can be readily prepared by carrying out the diazotization reaction under solvothermal conditions. The DAAQ molecules act as rigid support pillars to prevent the face-to-face stacking of the RGO flakes. More importantly, the active quinone carbonyl group of DAAQ provides additional pseudocapacitance. Therefore, the discharge capacitance of DAAQ-PGN is 522 F g^−1^ at a scanning rate of 5 mV s ^−1^.

#### 4.1.3. Acylation Reactions

Based on the development of organic molecular chemistry, the -NH_2_ of nucleophiles can react with carboxyl groups on conductive substrates (GO, MWCNTs, etc.). Acid groups are typically activated with thionyl chloride (SOCl_2_) [96,97,98], 1-ethyl-3-(3-dimethylaminopropyl)-carbodiimide (EDC) and *N*,*N*-dicyclohexylcarbodiimide (DCC) [99,100]. Subsequently, organic small molecules can be covalently modified on conductive substrates through amide bonds.

We decorated graphene hydrogels (RGO) with polyhydric organic small molecules (3,4,5-trihydroxybenzamide, THBA) through amidation reaction (Figure 9a) [80]. THBA acts both as an active center and prevents RGO sheet aggregation. Notably, the specific capacitance of RGO-THBA 1:3 at 5 mV s ^−1^ was 390.6 F g ^−1^, more than double that of RGO (175.5 F g ^−1^). This also confirms that the appropriate incorporation of THBA molecules is beneficial to decrease the equivalent series resistance and improve the specific capacitance value. All-carbon asymmetric supercapacitors (RGO-THBA//rGO-NDP) were constructed of RGO-THBA and rGO-1,4,5,8-naphthalenetetramethylenediimide (NDP), respectively. RGO-THBA//rGO-NDP has a capacitance of 70.8 F g ^−1^ and achieves an energy density of 14 W h kg ^−1^ at a power of 590 W kg ^−1^.

Kang et al. reported that 2,6-diaminopyridine (DAP) was partially anchored on the surface of RGO by an amide bond [81]. Another part of DAP molecules is embedded in RGO nanosheets through π–π interactions. The synthesized RGO/DAP nanocomposite with a hierarchical porous structure has a specific capacitance of 296.45 F g^−1^. The assembled RGO/DAP-1.5//AC device achieves a power density of 350 W kg^−1^ and an energy density of 14.55 Wh kg^−1^.

Likewise, Fan et al. used a chemical acylation strategy to prepare hollow carbon materials with heterogeneous interfaces (PPD-BC) [82]. In essence, the covalent bonding of p-phenylenediamine (PPD) to carbon spheres dramatically improves the overall properties of the material. Consequently, the PPD-BC shows the maximum specific capacitance of 451 F g^−1^ at the current density of 2 mV s^−1^ and superior cyclic stability (92%) over 5000 cycles. In addition, the adsorption energy between PPD molecules and the carbon surface for the covalent bridging model was calculated much higher than that in the non-covalent adsorption model. These results further validate that its ultra-long cycle life benefits from the stability of the covalent heterointerface.

In other respects, organic small-molecule covalently bound complexes have also played a prominent role in improving the electrical conductivity and stability of composites. Based on the acylation reaction, Elaheh Kowsari et al. functionalized graphene oxide (GO) by 5-aminopyridine-2-carboxylic acid Cr(Ⅲ) complex (FGO-Ap/Cr) using a one-pot hydrothermal method [83]. A high specific capacitance of 461.5 F g^−1^ is achieved at a current density of 1 A g^−1^ for the FGO-Ap/Cr in three electrodes electrochemical measurements. Among them, the increase in the number of multi-functional Ap/Cr ligands and accessible electroactive sites leads to a significant enhancement of the electrochemical behavior of FGO-Ap/Cr (Figure 9b,c).

#### 4.1.4. Cycloaddition Reaction

Among all covalent functionalization methods, 1,3-dipolar cycloaddition was first introduced by Huisgen in 1963 [101]. This reaction can take place under mild conditions and is applied to the functionalization of various carbon-based materials, such as fullerenes (C60s) [102], CNTs, and graphene [103,104,105]. Amit Paul’s group reacted 3,4-dihydroxybenzaldehyde with N-methyl glycine [84]. Subsequently, using pyridine as a base for proton abstraction to form 1,3-dipolar species (azomethine ylide). Finally, azomethine ylide was immobilized on graphene structures (HRFGs) via a 1,3-cycloaddition reaction with C=C double bonds. High conductivity, porosity, abundant redox-active groups, and stacked arrangement of graphene layers provide more accessible channels for electrolyte ion/electron transport. Figure 10a shows CV of HRFG at different scan rates in 2 M H_2_SO_4_. Two peaks were observed within 0.3–0.7 V, which could be attributed to the presence of redox-active oxygen functionalization on the graphene. The Pourbaix diagram has been plotted in Figure 10b for both peaks. Figure 10c,d show the GCD and the rate capability of HRFG at different current densities. The HRFGs single electrode exhibits a high capacitance (320 F g^−1^ at 1 A g^−1^; 389 F g^−1^ at 1 mV^−1^). As shown in Figure 10e, the Nyquist plot shows narrow semicircular features and vertical lines along the imaginary axis. The Bode plot and inset equivalent circuit show that HRFG has good capacitance characteristics.

In 2021, Rabah Boukherroub et al. reported a method for covalent functionalization of RGO via aryne cycloaddition with pseudocyclic iodoxoborole [85]. The strong covalent effect of the composite (f2-RGO) enhances the chemical/electrochemical stability of the attached functional groups compared to non-covalent methods. Furthermore, pseudocyclic iodoxoborole promoted the Faradaic reaction of the f2-RGO electrodes. According to the typical calculation formula, the specific capacitances of f2-RGO are 297 F g^−1^. The energy density of the assembled ASC can reach up to 6.7 W h kg^−1^ at a power density of 685.8 W kg^−1^.

#### 4.1.5. Doping of Heteroatoms

The energy storage performance of the composites can be improved by doping heteroatoms in the carbon framework through a covalent strategy. According to Tseng et al., different heteroatom doping affects the energy storage performance of conductive substrates [86]. First, the introduction of phytic acid (PA) as a phosphorus dopant provides graphene with abundant crosslinking sites. Subsequently, pentafluoropyridine organic compounds were used as fluorine- (F) and nitrogen (N)-doping precursors for the preparation of nitrogen, phosphorus, and fluorine co-doped graphene oxide (NPFG). In particular, the formed C-F covalent bonds exhibit ionic and semitonic properties. Benefiting from the high specific surface area and hierarchical pore structure, the NPFG-0.3 with the optimal ratio of heteroatomic weight exhibited a maximum capacitance of 319 F g^−1^ at 0.5 A g^−1^ in 6 M KOH. The symmetric supercapacitor device fabricated using NPFG-0.3 has a maximum specific energy of 38 Wh kg^−1^ and a maximum specific power of 716 W kg^−1^.

### 4.2. Non-Covalent Functionalization

Since most conductive carbon materials have large π bonds and 2D layered electronic structures, aromatic organic small molecules and conductive carbon can be combined by π–π interactions [106]. In addition, hydrogen bonding and electrostatic interactions were also used to modify the original carbon substrate surface with organic materials. This efficient approach ensures that the conductive sp^2^ carbon network and the high surface area of the electrodes are undisturbed.

#### 4.2.1. The π–π Interaction

π–π interaction is a special form of van der Waals force, which is a kind of supramolecular force. Since 2011, our group has reported a series of graphene-based organic molecule electrodes modified by different organic small molecules through π–π interactions. As a representative work, we prepared a novel hydrogel with both a 3D graphene framework and in-plane pores [25]. Subsequently, the redox-active caffeic acid (CFA) molecules were grafted onto the surface of novel hydrogels by non-covalent functionalization, denoted as G&GMH-CFA. The peak approximately +0.63 V of CV curve is attributed to the redox reaction of hydroxyl groups in CFA. Expanded ion transport channels and abundant active sites within the backbone provide G&GMH-CFA with a high specific capacitance of 482.6 F g^−1^ (at 1 A g^−1^). As shown in Figure 11 a–c, the optimum adsorption orientations of CFA molecules on the graphene were probed via First-principles calculations. Due to the enormous adsorption energy, the CFA molecular adsorption orientation parallel to graphene is the most stable mode.

Similarly, we also anchored 2,7-dihydroxy-9-fluorenone (DHFO) molecules on interconnected and highly conductive graphene sheets via non-covalent π–π interactions [87]. The π–π interactions between RGO and DHFO were beneficial for DHFO to grow and stack onto the RGO. Consequently, the DHFO/RGO hybrids achieved a high capacitance of 412.3 F g^−1^ (5 mV s^−1^) and great cycling durability. The factors affecting the redox potential of small organic molecules can be divided into changes in molecular skeleton and group positions or diversity of redox groups. On this basis, we selected π-conjugated aromatic 2,8-quinolinediol (QD) with heteroatom N as the active site donor [88]. This compound has one heteroatom N and two hydroxyl groups, enabling multiple electron transfers. Electrochemical tests show that the Faradaic reaction of the optimized OMEs (QD/RGO-0.75) occurs in a more positive potential range. Due to the respective advantages and synergistic effects of QD molecules and reduced graphene oxide, the QD/RGO-0.75 shows the maximum specific capacitance of 371 F g^−1^ at 5 mV s^−1^. Furthermore, our group designed and synthesized a novel quinone-like organic molecule (naphthalenediimine (NDI)) with a larger π-conjugated system [89]. NDI acts as a guest molecule for supercapacitors by non-covalently modifying rGO with high current responses in the potential range of −0.25 to −0.3 V. Benefiting from the contribution of redox charge reactions, the final rGO-NDI delivers a superior specific capacitance of 354 F g^−1^ at 5 mV s^−1^ and considerable cycling of ~87.2% after 8000 cycles.

#### 4.2.2. Hydrogen Bond Interactions

The presence of hydrophilic oxygen-containing functional groups (-OH, -O-, -COOH, etc.) enables the conductive substrate to interact with highly electronegative and small-radius atoms (F, O, N). Huai-Ping Cong et al. fabricated freestanding, transparent, and ultrathin graphene oxide films by sequentially depositing GO monolayers and melamine molecules [90]. The GO/melamine film all-solid-state flexible supercapacitors volumetric capacitance of 197.3 F cm^−3^ at a current density of 500 mA cm^−3^ due to the synergy of hydrogen bonds of GO/melamine and nitrogen atoms in the triazine ring. It is worth mentioning that this ultra-strong non-covalent interaction resulted in an ultimate strain of up to 3.5%.

In 2020, Chen used polydopamine (PDA) to modify the surface of activated carbon [91]. Subsequently, the amino and hydroxyl groups on the surface of the PDA film act as a multi-functional platform for the second reaction to interact with hydroquinone (HQ) via a wide range of hydrogen bonds (Figure 12). Electrochemical-related tests show that not only can the PDA undergo redox reactions in acidic media, but HQ can generate pseudocapacitance by gaining and losing two protons/electrons. Therefore, at 2 A g^−1^ C-PDA-1-3-HQ provides an excellent specific capacitance of 557 F g^−1^.

## 5. Conclusions and Perspectives

The development of new materials and devices for supercapacitors is exciting. The discovery and creation of new materials with tailored architectures offer a wonderful opportunity to increase power and energy density. Therefore, the design of novel SCs materials should focus on enhancing capacitance by incorporating redox motifs into conventional EDLC. Various organic small-molecule electrode materials have been recently developed due to resource renewability, environmental friendliness, and structural diversity. This paper reviews the research progress of some representative organic small-molecule electrodes in the application of supercapacitors. The tunable active groups enable them to exhibit different energy storage mechanisms and electrochemical performances. In particular, quinones and their derivatives have fast charge/discharge rates and relatively high energy densities as redox-active compounds. On this basis, electron-withdrawing/donating groups on the molecular backbone can affect the electron affinity of OMEs, thus showing tunable redox potentials. In addition, some strategies were developed during the preparation of OMEs to further enhance their electrochemical performance, such as covalent functionalization and non-covalent functionalization. The covalent functionalization of conductive substrates by organic small molecules is accompanied by the rehybridization of C atoms from sp^2^ to sp^3^. The formation of covalent bonds can reduce the solubility of organic small molecules and ensure the high stability of supercapacitors. The non-covalent interaction ensures that the sp^2^ carbon network of the conductive substrate is not disturbed while maintaining the high surface area and redox activity of the electrode. Moreover, when the mirror-symmetrical redox peak of OMEs is present in the CV curve, the assembled supercapacitors showed good self-coordination and potential self-matching behavior.

Although some OMEs show great promise in energy storage, several technical hurdles must be resolved before their practical application. Firstly, to further improve the electrochemical performance of organic small-molecule electrodes, it is necessary to understand the detailed redox processes, such as molecular structure evolution (intermediates) and charge storage mechanism. In fact, most OMEs exhibit surface-controlled capacitance behavior based on the reaction kinetics. Therefore, it is crucial to develop strategies to increase the contribution of pseudocapacitance in redox processes to improve the rate performance. On the other hand, the electrochemical performance of OMEs depends on their molecular structure. As discussed above, organic small molecules can reasonably design by adding active and electron absorption/donor groups. In addition, in situ testing, volatile intermediates in the redox process of OMEs can be monitored and captured. Secondly, OME materials exhibit excellent electronic conductivity and enhanced insolubility due to the composite involving organic small molecules and carbon substrates. However, the microstructure of OMEs is usually disordered and random. Therefore, it can be considered to design OMEs with controllable morphology by adjusting the reaction kinetics. At the molecular level, organic small molecules with reversible redox activity can be integrated into conjugated microporous polymers (CMP) and covalent organic frameworks (COF) to further combine with conductive substrates. With an in-depth understanding of intramolecular and intermolecular interactions, organic small-molecule electrodes are considered the most promising alternative for efficient energy storage.

## Data Availability

Not applicable.

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
