# Peer review of "Organic Small-Molecule Electrodes: Emerging Organic Composite Materials in Supercapacitors for Efficient Energy Storage"

_molecules, 2022, doi:10.3390/molecules27227692_

Round 1

Reviewer 1 Report

In this paper, the authors introduced the latest development of organic molecular electrodes for efficient energy storage. Furthermore, we focus on the design motivation, structural advantages, charge storage mechanism and various electrode parameters of organic molecular electrodes. The reviewer suggests that the manuscript could be accepted for publication after the revisions as following:

1.       Some images are blurry, such as Figure 6a, et al. Higher quality should be reorganized.

2.       I would like the authors add some comments about the Supercapacitors to enrich this work, such as: Honeycomb-like nitrogen-doped porous carbon decorated with Co3O4 nanoparticles for superior electrochemical performance pseudo-capacitive lithium storage and supercapacitors; Recent progress in cathode catalyst for nonaqueous lithium oxygen batteries: a review; Embedding NiS nanoflakes in electrospun carbon fibers containing NiS nanoparticles for hybrid supercapacitors, and Morphology controlled hierarchical NiS/carbon hexahedrons derived from nitrilotriacetic acid-assembly strategy for high-performance hybrid supercapacitors.

3.       The electrochemical performance of recent studies about several categories of Organic Molecular Electrodes are suggested to be summarized and listed in a Table in this work.

4.       More discussion about prospect in the last section of this review should be added to offer the guidance for further exploration. 

5.       The manuscript contains spelling/grammatical errors. So, the language should be polished thoroughly.

Reviewer 2 Report

The manuscript reported progress on organic small molecular composite electrodes and their capacitive performance. This review discusses the design motivation, structural advantages, charge storage mechanism, classification and preparation methods of organic electrodes. The contents were comprehensive and detailed, and the expression was clear. And the manuscript has a good significance. However, there were some problems in the manuscript, thus the acceptance after major revisions is recommended.

1.       There are many types of organic electrodes, and this review focuses on organic small molecular composite electrodes, so it is recommended that the title be revised to “Organic Small Molecular Electrodes: Emerging Organic Composite Materials in Supercapacitors for Efficient Energy Storage” to focus on the thrust of this review.

2.       Some reviews have discussed the application of small organic molecules in supercapacitors (such as DOI: 10.3390/ma12111770, 10.1039/c6cs00173d). Please highlight the innovation of this review.

3.       To better introduce the supercapacitor in introduction, the following reference are suggested to be included: 10.1007/s42773-022-00176-9; 10.1016/j.apsusc.2022.155144.

4.       Remove the incomplete and unnecessary outlines in the Figures. In addition, the contents of the existing Figures are relatively monotonous, and it is suggested to supplement some other types of Figures, such as the SEM and/or TEM images of the prepared organic small molecule electrodes.

5.       Line 139, what is the electrochemical catalysis of carbon substrates?

6.       It is suggested to introduce the development history of the organic electrode materials in supercapacitors.

7.       Please check the title of each section. “2. Classification of Organic Molecular Electrodes” should be changed to “3. Classification of Organic Molecular Electrodes”. “3.1.2. Quinone Derivatives” should be changed to “3.1.2. Ketones”. “4.1.4. Doping of Heteroatoms” should be changed to “4.1.5. Doping of Heteroatoms”.

8.       Please explain the effect of the proportion of organic small molecular in the composites on the capacitance properties.

9.       The electrolyte used in most studies is 1 M H2SO4, which is related to the H+ required for the redox reaction of the active functional groups.  But neutral electrolytes are used in a few studies (line 417). Why? Can electrodes fully exploit their advantages in neutral electrolytes?

10.    Line 506, “Figure 10 shows the electrochemical correlation test of HRFGs”, such statements are too general.

11.    Please add Tables to present synthetic views of the literature and systems, develop comprehensive assessments. Please refer and cite 10.1039/D2NR01986H.

Round 2

Reviewer 1 Report

The author has answered the reviewer's information well,I suggeste that the manuscript can be accept for publication now

Reviewer 2 Report

The manuscript has been well revised and could be accepted.